# OpenReview forum: "Mitigating Fine-tuning Risks in LLMs via Safety-Aware Probing Optimization"
_ICLR.cc/2026/Conference — Submitted to ICLR 2026_

### Official Review · Reviewer_PBzG · 2025-10-31

**Soundness:** 2
**Presentation:** 3
**Contribution:** 2
**Rating:** 4
**Confidence:** 5

**Summary:**

The paper solves the LLM safety degradation caused by fine-tuning. It hypothesizes that usefulness-critical and safety-critical gradient directions are entangled and proposes Safety-Aware Probing (SAP).  At each step, SAP estimates a safety-critical direction using a contrastive safety loss, constructs a “harmful” update, and then learn a small hidden-state probe that maximizes a safe-useful loss, encouraging downstream weight updates to avoid harmful regions. Across three 7b models on benign and adversarial tasks, SAP reduces harmfulness scores while keeping task loss/metrics near SFT. Additional experiments demonstrate the effectiveness of different settings. However, the scalability is not validated and remains questionable.

**Strengths:**

1. The proposed SAP is well motivated. The hypothesis is validated by experiments.
2. Experiments over 3 models and several datasets in different settings demonstrate promising performance.
3. It provides sensible ablations (e.g., which layers to probe and learning-rate sensitivity) and report costs (time and memory).

**Weaknesses:**

1. The first claimed contribution of validating the hypothesis has been explored in prior works, such as [1, 2]. The cosine similarity approach is similar to SafeLora [1].

2. The authors claim in the introduction that their work “*has better scalability since it can be incorporated into various fine-tuning paradigms rather than being limited to LoRA.*”  However, this claim is not supported by experimental evidence, such as larger models or fully fine-tuning. All experiments are conducted on 7B models with LoRA.  Furthermore, this claim is questionable, as SAP requires extra gradient estimation, which is related to the number of trainable parameters.

3. The time overhead of SAP is non-trivial, approximately 2.5 times that of all baselines. While Appendix B.5 includes a comparison with LISA, the discussion does not adequately address the overhead relative to other baselines. Considering the performance gain, the practical value of applying this method remains questionable.

4. Figure 1 lacks clarity and should be improved to better convey the procedure of SAP.


**Reference**

[1]  Safelora: The silver lining of reducing safety risks when finetuning large language models. NeurIPS. 2024

[2] Safe Delta: Consistently Preserving Safety when Fine-Tuning LLMs on Diverse Datasets. ICML. 2025

**Questions:**

1. What is the time cost for SAP when applying to larger models and fully fine-tuning settings?
2. What is the portion or number of safety examples for baselines, such as safeinstr?
3. Why not use fewer, smaller probe layers instead of probing 10 layers, given the high time cost?

---

> ### Author Response · Authors · 2025-11-21
> **Response to Reviewer PBzG (Part 1/3)**
>
> Dear Reviewer PBzG,
>
> Thank you for your insightful feedback and recognizing that our work is well motivated, demonstrating promising performance, and providing sensible ablations. In response to your concerns, we have carefully revised the manuscript and provide detailed responses below.
>
> ---
>
> **Q1**: What is the time cost for SAP when applying to larger models and fully fine-tuning settings?
>
> **A1**: Thanks for the insightful question. We additionally conducted experiments of SAP under larger model (Llama-2-13B) or full fine-tuning (Qwen-0.6B due to computational limitations), where the computational profile of our method on Qwen-0.5B and Llama2-13B is shown below. These results demonstrate that the time cost of SAP is still admissible under different settings.
>
> | Model | llama2-7B |  | qwen-0.6B |  | llama2-13B |  |
> | --- | --- | --- | --- | --- | --- | --- |
> |  | SFT | SAP | SFT | SAP | SFT | SAP |
> | Clock time per batch (s) | 0.38 | 1.09 | 0.27 | 0.62 | 0.85 | 2.14 |
> | GPU Memory (GB) | 40.81 | 40.87 | 12.45 | 12.63 | 72.97 | 73.58 |
>
> ---
>
> **Q2**: What is the portion or number of safety examples for baselines, such as safeinstr?
>
> **A2**: As detailed in Appendix C.3 (More Details for Baselines), methods that incorporate safety data (e.g., Lisa and SafeInstr) utilize 200 and 150 samples, respectively. In contrast, our approach requires only 50 safety samples, achieving superior safety performance with significantly less data.
>
> ---
>
> **Q3**: Why not use fewer, smaller probe layers instead of probing 10 layers, given the high time cost?
>
> **A3**: We thank the reviewer for this valuable suggestion. Our experiments in Appendix B.7 (Probe Layer Variability Analysis) indeed show that even randomly probing only 2 layers can achieve safety performance comparable to using 10 layers, confirming the potential of smaller probe sets. Overall, the comprehensive experiments across randomly probing 10 layers (Table 7, Section 5.4) or 2 layers (Table 12, Appendix B.7) validate that **the selection of probing layers does not significantly impact the overall result of SAP**, and we suggest using a few layers (per computational budget) in the middle as the default, based on empirical observations on these results.

---

> ### Author Response · Authors · 2025-11-21
> **Response to Reviewer PBzG (Part 2/3)**
>
> **Q4**: The first claimed contribution of validating the hypothesis has been explored in prior works, such as [1, 2]. The cosine similarity approach is similar to SafeLora [1].
>
> **A4**: Thanks for your insightful comment. We have explicitly cited [1,2] in our introduction to acknowledge their conceptual similarities, and we also kindly note that a key difference between our work and [1,2] is that we are the first to provide a **qualitative analysis of the entangled gradient directions** (useful-critical and safety-critical), unlike the cosine similarity threshold proposed by SafeLora **in the parameter space**. Thank you once again for bringing these two works to our attention, and we have carefully revised our introduction based on the discussion above.
>
> [1] Safelora: The silver lining of reducing safety risks when finetuning large language models. NeurIPS. 2024
>
> [2] Safe Delta: Consistently Preserving Safety when Fine-Tuning LLMs on Diverse Datasets. ICML. 2025
>
> ---
>
> **Q5**: The authors claim in the introduction that their work “*has better scalability since it can be incorporated into various fine-tuning paradigms rather than being limited to LoRA.*” However, this claim is not supported by experimental evidence, such as larger models or fully fine-tuning. All experiments are conducted on 7B models with LoRA. Furthermore, this claim is questionable, as SAP requires extra gradient estimation, which is related to the number of trainable parameters.
>
> **A5**: Thank you for the kind suggestion. First, we would like to clarify that the *scalability* here refers to its adaption to different fine-tuning paradigms, including full-parameter fine-tuning and LoRA, etc, not the scalability on model size. Following your comment, we have conducted additional experiments including full-parameter fine-tuning on the Qwen3-0.6B model and LoRA fine-tuning on the Llama2-13B model (with the same setup as in A1). The results demonstrate the consistent effectiveness of our proposed method.
>
> | Model | Qwen-0.6B |  |  | Llama2-13B |  |  |
> | --- | --- | --- | --- | --- | --- | --- |
> | Method | BRT | CL | HS | BRT | CL | HS |
> | Origin | 0.469 | 11.42 | 36.10 | 0.462 | 17.33 | 29.20 |
> | LoRA | 0.516 | 5.44 | 39.80 | 0.517 | 5.42 | 34.40 |
> | SAFT | 0.499 | 5.56 | 30.80 | 0.495 | 5.51 | 30.70 |
> | Lisa | 0.498 | 5.69 | 30.20 | 0.504 | 5.56 | 24.80 |
> | safeInstr | 0.505 | 5.64 | 28.60 | 0.515 | 5.48 | 28.30 |
> | SaLoRA | 0.511 | 5.59 | 30.50 | 0.512 | 5.56 | 28.90 |
> | Ours | 0.518 | 5.47 | 25.50 | 0.525 | 5.44 | 22.10 |
>
> In terms of fine-tuning performance, our approach achieves nearly identical BRT scores and Cross-Entropy (CL) loss compared to Standard Fine-Tuning (SFT), confirming that it does not compromise the model's primary task learning capability. Crucially, for the Qwen3-0.6B model, our method reduces the harmfulness score by 14.3 points compared to SFT; for the Llama2-13B model, it achieves a reduction of 12.3 points in harmfulness score compared to SFT, highlighting its significant enhancement of model safety. Furthermore, compared to other baseline methods, our approach achieves a Pareto improvement, excelling in both fine-tuning performance and safety metrics.

---

> ### Author Response · Authors · 2025-11-21
> **Response to Reviewer PBzG (Part 3/3)**
>
> **Q6**: The time overhead of SAP is non-trivial, approximately 2.5 times that of all baselines. While Appendix B.5 includes a comparison with LISA, the discussion does not adequately address the overhead relative to other baselines. Considering the performance gain, the practical value of applying this method remains questionable.
>
> **A6**: Thanks for the thoughtful comment. The reason why we only compare SAP with LISA in the previous version is that LISA is a strong baseline. However, we have also conducted similar experiments for other methods, demonstrating that our approach maintains optimal performance even with the same fine-tuning time, and the result is as follows:
> previous version is that LISA is a strong baseline.
>
> |  | Ours |  |  | LoRA |  |  | SAFT |  |  |
> | --- | --- | --- | --- | --- | --- | --- | --- | --- | --- |
> | epoch | BRT | CL | HS | BRT | CL | HS | BRT | CL | HS |
> | 1 | 6.89 | 0.462 | 28.5 | 7.02 | 0.457 | 31.5 | 7.07 | 0.458 | 31.3 |
> | 2 | 6.56 | 0.474 | 26.4 | 6.69 | 0.465 | 31.7 | 6.72 | 0.462 | 31.6 |
> | 3 | 6.28 | 0.481 | 24.9 | 6.41 | 0.473 | 31.6 | 6.58 | 0.469 | 32.1 |
> | 4 | 6.13 | 0.495 | 23.1 | 6.23 | 0.488 | 32.5 | 6.32 | 0.482 | 31.8 |
> | 5 | 6.07 | 0.518 | 22.8 | 6.17 | 0.493 | 32.3 | 6.27 | 0.487 | 31.4 |
> | 6 |  |  |  | 6.14 | 0.507 | 32.8 | 6.24 | 0.491 | 31.6 |
> | 7 |  |  |  | 6.11 | 0.505 | 33.3 | 6.21 | 0.490 | 31.2 |
> | 8 |  |  |  | 6.09 | 0.511 | 33.1 | 6.14 | 0.505 | 31.7 |
> | 9 |  |  |  | 6.10 | 0.513 | 34.2 | 6.12 | 0.506 | 32.5 |
> | 10 |  |  |  | 6.07 | 0.517 | 33.8 | 6.13 | 0.502 | 32.3 |
> | 11 |  |  |  | 6.05 | 0.519 | 34.1 | 6.10 | 0.516 | 32.7 |
> | 12 |  |  |  | 6.03 | 0.516 | 33.9 | 6.12 | 0.515 | 32.4 |
> | 13 |  |  |  | 6.06 | 0.514 | 34.5 | 6.14 | 0.518 | 32.1 |
> | 14 |  |  |  | 6.04 | 0.518 | 34.2 | 6.11 | 0.516 | 32.8 |
> | 15 |  |  |  | 6.03 | 0.517 | 34.6 | 6.13 | 0.517 | 32.5 |
> |  |  |  |  |  |  |  |  |  |  |
> |  | Lisa |  |  | safeInstr |  |  | SaLoRA |  |  |
> | epoch | BRT | CL | HS | BRT | CL | HS | BRT | CL | HS |
> | 1 | 8.11 | 0.456 | 29.1 | 7.05 | 0.452 | 30.5 | 7.63 | 0.451 | 30.7 |
> | 2 | 7.49 | 0.458 | 28.4 | 6.72 | 0.459 | 31.2 | 7.02 | 0.462 | 31.2 |
> | 3 | 6.85 | 0.465 | 27.7 | 6.51 | 0.468 | 30.7 | 6.64 | 0.465 | 31.5 |
> | 4 | 6.53 | 0.476 | 27.1 | 6.35 | 0.473 | 30.1 | 6.52 | 0.471 | 30.9 |
> | 5 | 6.37 | 0.481 | 26.8 | 6.22 | 0.481 | 29.5 | 6.47 | 0.475 | 30.5 |
> | 6 | 6.29 | 0.482 | 26.4 | 6.16 | 0.495 | 29.3 | 6.41 | 0.482 | 30.2 |
> | 7 | 6.22 | 0.489 | 25.9 | 6.14 | 0.504 | 29.7 | 6.38 | 0.478 | 29.8 |
> | 8 | 6.16 | 0.492 | 25.7 | 6.13 | 0.508 | 28.8 | 6.32 | 0.475 | 29.5 |
> | 9 | 6.14 | 0.495 | 25.5 | 6.11 | 0.511 | 28.5 | 6.25 | 0.484 | 29.7 |
> | 10 | 6.15 | 0.493 | 25.6 | 6.08 | 0.512 | 28.4 | 6.21 | 0.487 | 29.3 |
> | 11 | 6.12 | 0.499 | 25.2 | 6.06 | 0.516 | 28.2 | 6.15 | 0.492 | 29.4 |
> | 12 | 6.13 | 0.495 | 25.4 | 6.08 | 0.517 | 28.4 | 6.14 | 0.497 | 29.5 |
> | 13 | 6.11 | 0.503 | 24.9 | 6.08 | 0.514 | 28.3 | 6.11 | 0.508 | 29.2 |
> | 14 | 6.14 | 0.497 | 25.1 | 6.07 | 0.515 | 28.6 | 6.13 | 0.505 | 29.1 |
> | 15 | 6.12 | 0.501 | 24.9 | 6.06 | 0.516 | 28.1 | 6.14 | 0.506 | 29.2 |
>
> As shown, our method achieves performance superior to most baselines with only 5 epochs. In terms of fine-tuning performance, except for the CL loss of the LoRA method after 11 epochs being lower than ours, all other baselines are outperformed by SAP in both CL loss and BRT score. More importantly, with only 5 epochs, SAP reduces the HS to 22.8—a safety level that other baselines struggle to reach even after 15 epochs. This demonstrates that under equal fine-tuning time constraints, our method significantly surpasses the baselines in both fine-tuning capability and safety performance.
>
> ---
>
> **Q7**: Figure 1 lacks clarity and should be improved to better convey the procedure of SAP.
>
> **A7**: Thank you for your suggestion. The goal of Figure 1 is to provide a high-level overview of SAP, not the detailed procedure, which can be referred to Section 4 and Algorithm 1. However, to better help readers better understand its underlying mechanism, we have added more detailed explanations for the concepts shown in Figure 1 in its caption:
>
> > A brief overview of SAP (b) and its comparison with standard fine-tuning (a). The key design of SAP lies in perturbing the hidden state with safety-critical directions, which assists in eluding potentially harmful regions during optimization in advance. Specifically, as described in Algorithm 1, we employ safety data to gain a safety correlated parameter direction $\nabla_WL_\text{safety}$ (described in Equation 3),  which is used to compute a safety hidden state probe $V_\text{safe}$ in model parameters to ensure a safe fine-tuning.
>
> We hope this additional clarification could help better convey the mechanism of SAP.
>
> ---
>
> Thank you once again for your constructive feedback. If you have any questions or concerns, please let us know.

---

> > ### Author Response · Authors · 2025-11-27
> >
> > Dear Reviewer PBzG,
> >
> > It is our great honor to have you as a reviewer for our paper. We sincerely appreciate the time and effort you have devoted to your review. We have carefully clarified or addressed all the concerns you raised, and sincerely hope that these clarifications and revisions have resolved them.
> >
> > As the discussion phase is nearing its end, we just wanted to gently check whether you have any remaining concerns. We would be very happy to clarify anything further if needed.
> >
> > Thank you again for your thoughtful feedback. Wishing you a lovely day!
> >
> > Warmly,
> >
> > Authors

---

### Official Review · Reviewer_kkBd · 2025-11-03

**Soundness:** 2
**Presentation:** 2
**Contribution:** 3
**Rating:** 6
**Confidence:** 3

**Summary:**

This paper shows that large language models can lose their safety even when fine-tuned on harmless data, because the gradients that improve task performance are often entangled with those that reduce safety. To address this, the authors propose Safety-Aware Probing (SAP), a training method that adds a small hidden-state probe to steer optimization away from harmful directions while still improving task performance. SAP does not require changing the dataset or model architecture and works across different fine-tuning setups.

**Strengths:**

The experiments are extensive and well designed, have evaluations on three different models, three instruction-following datasets, five reasoning benchmarks, and poisoned, adversarial fine-tuning settings.
The paper is also clearly written and well-organized.

**Weaknesses:**

* Gradient analysis and evaluation are somewhat limited. The paper shows cosine similarity between usefulness and safety gradients, but does not fully explain why the directions align or provide deeper theoretical insight. The Harmful score relies on a single moderation model, with no additional metrics such as jailbreak success rate, or LLM-as-judge evaluation.
* Cost analysis is needed. SAP increases training time by 2x~3x, but the paper briefly labels this as acceptable without discussing practical implications or scalability.
* Limited discussion of adaptive attacks. The adversarial fine-tuning experiment does not consider attackers aware of SAP.

**Questions:**

NA

---

> ### Author Response · Authors · 2025-11-21
> **Response to Reviewer kkBd**
>
> Dear Reviewer kkBd,
>
> Thank you for your insightful feedback and recognizing that our experiments are extensive and well-designed, our paper clearly written and well-organized. In response to your concerns, we have carefully revised the manuscript and provide detailed responses below.
>
> ---
>
> **Q1**: Gradient analysis and evaluation are somewhat limited. The paper shows cosine similarity between usefulness and safety gradients, but does not fully explain why the directions align or provide deeper theoretical insight. The Harmful score relies on a single moderation model, with no additional metrics such as jailbreak success rate, or LLM-as-judge evaluation.
>
> **A1**: Thank you for your insightful comment. We agree that a more thorough theoretical analysis of the alignment between usefulness and safety gradients would strengthen our claims. In our work, the cosine similarity measurements (Fig. 3) and the dynamics of $L_{su}$ (Fig. 6) serve as initial empirical evidence supporting our hypothesis of gradient entanglement, which aimed to provide the motivation of SAP. Regarding theoretical analysis, while we acknowledge that further theoretical grounding would be valuable, we respectfully note that this is out of the scope of our paper, and our main contribution lies in the algorithmic design and empirical validation of SAP.
>
> Regarding safety evaluation, we employed the BeaverTails moderation model, which is **exactly an LLM-as-judge evaluation** metric (the model is fine-tuned from a base LLM). This model has been widely recognized and authoritative for assessing the harmfulness of model outputs. Additionally, the Harmful Score (HS) is a numerical rating (ranging from 1-5) of jailbreaking harmfulness, providing a more robust assessment than jailbreak success rate, thus we consider Harmful Score as our core evaluation metric. Overall, we appreciate the reviewer's valuable suggestions on evaluation metrics, but we kindly note that current metrics have already provided a comprehensive assessment of overall safety performance.
>
> ---
>
> **Q2**: Cost analysis is needed. SAP increases training time by 2x~3x, but the paper briefly labels this as acceptable without discussing practical implications or scalability.
>
> **A2**: We appreciate the reviewer’s attention to computational cost. It is true that SAP introduces additional computational overhead, approximately doubling the training time compared to standard fine-tuning, as shown in Table 6, due to the bi-level optimization process involving both $W$ and $V$ updates. However, we note that the GPU memory footprint remains similar to SFT, making it feasible on typical hardware setups.
> ﻿Regarding the training time, we have provided an in-depth comparison under the same total training time in **Figure 5**, where we compare SAP (with 5 epochs) and a strong baseline, Lisa (with 15 epochs). Under this comparison, the total training time is kept the same, but SAP still outperforms Lisa across all metrics (both safety and utility), indicating better efficiency of SAP even with fewer training time.
>
> ---
>
> **Q3**: Limited discussion of adaptive attacks. The adversarial fine-tuning experiment does not consider attackers aware of SAP.
>
> **A3**: We thank the reviewer for this suggestion. We would like to highlight that we have already conducted preliminary experiments on adaptive attacks in Appendix B.4 (Consideration for Adaptive Attacks), where the attacker is assumed to be aware of the SAP defense and attempts to perform targeted attacks. Specifically, we consider modifications that can decrease the model safety by steering the fine-tuning away from the safe region, targeting the safe parameter region pursued by SAP. This attack can be implemented before or after the application of SAP, formulating two adaptive attack scenarios:
>
> - **Pre-Fine-Tuning Attack**. Under this attack, the model is compromised for 5 epochs before the SAP/SFT application. Still, we use Alpaca as the useful dataset with default hyperparameters for SAP/SFT.
> - **Post-Fine-Tuning Attack**. In this setting, the attacker directly attacks an SAP-fine-tuned model for 10 epochs. During the fine-tuning phase, we use Alpaca as the useful dataset with default hyperparameters for SAP/SFT.
>
> As shown in Table 9 and Figure 9, SAP maintains strong safety performance even under such adaptive settings. Specifically, when the attacker conducts attacks before fine-tuning, our method reduces the Harmful Score by 12.7 compared to SFT. When attacks are performed after fine-tuning, SAP achieves a reduction in Harmful Score of approximately 33 points after 10 attack steps. These results demonstrate that SAP exhibits considerable robustness even when the adversary adapts specifically to circumvent our defense.
>
> ---
>
> Thank you once again for your constructive feedback. If you have any questions or concerns, please let us know.

---

### Official Review · Reviewer_3MFr · 2025-11-03

**Soundness:** 2
**Presentation:** 2
**Contribution:** 2
**Rating:** 4
**Confidence:** 3

**Summary:**

The paper introduces Safety-Aware Probing (SAP), a lightweight optimization that inserts a small safety-aware probe into hidden states during gradient propagation to steer updates away from harmful directions while preserving task utility. SAP is motivated by an observed entanglement between safety-critical and usefulness-critical gradient directions; it maximizes a safe-useful objective to find a probe  that biases each update toward safer regions.

**Strengths:**

- The proposed method is simple yet principled: it treats safety as gradient-space steering and remains broadly compatible with standard fine-tuning.
- Empirically, it improves utility while reducing harmfulness, increases robustness to poisoning and adversarial fine-tuning, and composes with other defenses, making it practical for deployment.

**Weaknesses:**

The proposed method introduces too many hyperparameters (e.g., α, β, ϵ, probe layers), which increases tuning complexity and reduces reproducibility.

**Questions:**

Q1. Did you mean the following?

“Our experiments show that SAP achieves better useful loss while significantly **decreasing model safety”**

→ “Our experiments show that SAP achieves better useful loss while significantly **improving model safety**”

Q2. Could you clarify the captions for Figures 2 and 3?

- Figure 2 does not specify which harmful or useful
datasets were used.
- Figure 3 does not clarify the definition of the useful-critical notation nor specify which harmful dataset was used.

Q3. Are $\alpha$, $\beta$, $\epsilon$, and probe layers the same for every dataset?

Q4. Is there a reason for choosing 2,000 examples for $D_{useful}$ and 50 for $D_{safe/harmful}$?

Q5. Aren’t these models instruction-tuned models, such as Llama-7B-Chat?

Q6. Why are the Booster and Vaccine baselines not included in the results after Table 1?

Formatting issue:

- The repeated inclusion of the Alpaca dataset in Table 2 is redundant.

---

> ### Author Response · Authors · 2025-11-21
> **Response to Reviewer 3MFr (Part 1/2)**
>
> ### Reviewer 3MFr
>
> Dear Reviewer 3MFr,
>
> Thank you for your insightful feedback and recognizing our work as simple yet principled and practical for deployment. In response to your concerns, we have carefully revised the manuscript and provide detailed responses below.
>
> ---
>
> **Q1**. Did you mean the following?
>
> “Our experiments show that SAP achieves better useful loss while significantly **decreasing model safety”**
>
> → “Our experiments show that SAP achieves better useful loss while significantly **improving model safety**”
>
> **A1**: Thank you for your careful reading! This is a typo, and we have fixed it.
>
> ---
>
> **Q2**.Could you clarify the captions for Figures 2 and 3?
>
> - Figure 2 does not specify which harmful or useful datasets were used.
> - Figure 3 does not clarify the definition of the useful-critical notation nor specify which harmful dataset was used.
>
> **A2**: Thank you for raising this point. Originally, we stated that “More details on the calculation of $L_\text{safety}$ are illustrated in Section 5”. To make these settings more accessible, we have added the details in the captions. Specifically:
>
> - In Figure 2, we use alpaca (useful) and CircuiteBreaker (harmful) dataset, which are the same as the default setting in Section 5.
> - In Figure 3, the useful-critical direction is the gradient direction for the fine-tuning task, which is defined in Equation (5), Definition 3.2 in Section 3.2. The datasets are kept the same as Figure 2.
>
> ---
>
> **Q3 & W1**: Are $\alpha$, $\beta$, $\epsilon$ and probe layers the same for every dataset?
>
> The proposed method introduces too many hyperparameters (e.g., $\alpha$, $\beta$, $\epsilon$, probe layers), which increases tuning complexity and reduces reproducibility.
>
> **A3**: Thank you for raising this point. First, we clarify that these hyperparameters are the same for every dataset in our main experiments (Table1-5), as stated in Section 5.1 (paragraph *general configurations for SAP*):
>
> > The update steps (learning rate) for $W$, $V$, and $\Delta W_\text{harmful}$ are $\alpha=$1e-4, $\beta=$5e-2, and $\epsilon=$2e-5, respectively.
>
> While our method introduces some hyperparameters, they are either robust against selection or can be adaptively set. Specifically:
>
> - $\alpha$: This is the parameter update step size for LLM, a.k.a. **the learning rate** of the fine-tuning LLMs. Thus, $\alpha$ does not introduce new hyperparameters during fine-tuning. Additionally, We have conducted experiments in Table 8, Section 5.4 to demonstrate the robustness of SAP against the selection of $\alpha$.
> - $\beta$: This is the update step size for the probe $V$. To automatically set this parameter, we have proposed an adaptive variant of SAP that dynamically calibrates $\beta$ during fine-tuning, which is detailed in Appendix B.5. The proposed variant (adaptive SAP) performs comparably with SAP under carefully selected $\beta$, showing the effectiveness of the adaptive $\beta$ schedule.
> - $\epsilon$: This parameter is only set to find the harmful direction $\Delta W_\text{harmful}$, which intuitively does not affect the perturbation performance. We kept it as $\epsilon=2e-5$ across all models and datasets, which is universal and does not require careful selection.
> - Probe layers: We conducted experiments to demonstrate the robustness of SAP even under random probe layer selection, as detailed in Table 3, Section 5.4, and Table 12, Appendix B.7. These experiments validate that the effectiveness of SAP is universal across different layer selections.

---

> ### Author Response · Authors · 2025-11-21
> **Response to Reviewer 3MFr (Part 2/2)**
>
> **Q4**:  Is there a reason for choosing 2,000 examples for $D_{useful}$ and 50 for $D_{safe/harmful}$
>
> **A4**: Thanks for your careful consideration. For $D_{useful}$, which is exactly the fine-tuning task dataset, we keep it the same as the original task, and its number does not affect the performance of SAP. For $D_{safe/harmful}$, since we only need to find safety-critical directions with them, this process generally does not require a lot of safety-related data. We also conducted an evaluation regarding the selection of $D_{safe/harmful}$ in Table 13, Appendix B.8, where the sampling of this dataset does not significantly affect the results.
>
> ---
>
> **Q5**. Aren’t these models instruction-tuned models, such as Llama-7B-Chat?
>
> **A5**: Yes, the models we evaluated are instruction-tuned. However, instruction-tuned models still suffer from compromising safety after task-specific fine-tuning [1], making this safety issue unresolved. To mitigate this risk, our work proposes SAP to preserve safety after fine-tuning, including instruction-tuned models.
>
> [1] Fine-tuning Aligned Language Models Compromises Safety, Even When Users Do Not Intend To!. ICLR 2024
>
> ---
>
> **Q6**: Why are the Booster and Vaccine baselines not included in the results after Table 1?
>
> **A6**: Actually, Booster and Vaccine are designed for general alignment enhancement, not for preserving safety during task-specific fine-tuning. By contrast, SAP and other baselines like SAFT, Lisa are designed for the latter goal. Thus, we mainly compare SAP with baselines from the same setting, while also considering methods like Booster and Vaccine from other settings. To make this point clearer, we have explained this difference in our experiment setup, and additionally conducted new experiments in Table 2 (copied below):
>
> |  | Alpaca |  |  | Samsum |  |  | Chatdoctor |  |  | Average |  |  |
> | --- | --- | --- | --- | --- | --- | --- | --- | --- | --- | --- | --- | --- |
> |  | BRT | CL | HS | BRT | CL | HS | BRT | CL | HS | BRT | CL | HS |
> | SFT | 0.514  | 6.06 | 33.1 | 0.541  | 1.79  | 35.40  | 0.464  | 6.16  | 27.30  | 0.506  | 4.67  | 31.93  |
> | Booster | 0.494 | 6.09 | 27.90 | 0.536 | 1.87 | 26.30 | 0.461 | 6.19 | 24.40 | 0.497 | 4.72 | 26.20 |
> | Vaccine | 0.487 | 6.13 | 28.50 | 0.527 | 1.94 | 27.20 | 0.458 | 6.22 | 26.10 | 0.491 | 4.76 | 27.27 |
> | SAP (Ours) | **0.521** | **6.03** | **22.60** | **0.539** | **1.75** | **21.70** | **0.463** | **6.15** | **20.80** | **0.508** | **4.64** | **21.70** |
>
> In the experiments presented in the table we can observe that our method outperforms both baselines on the fine-tuning dataset, achieving an average BRT score improvement of 0.011 and 0.017 compared to Booster and Vaccine, respectively. Meanwhile, the average CL loss is reduced by 0.08 and 0.12 relative to Booster and Vaccine. More importantly, our method significantly lowers the Harmful Score by 4.50 and 5.57 compared to Booster and Vaccine, respectively. This marked reduction underscores the enhanced safety assurance of our proposed approach.
>
> We have added these results to Table 2 and the related discussion in Section 5.
>
> ---
>
> **Q7**: The repeated inclusion of the Alpaca dataset in Table 2 is redundant.
>
> **A7**: Thank you for your careful reading. While redundant, the results in Table 1 and Table 2 are presented to compare results across different models and datasets, and more importantly, we need to calculate the average score for these two comparison dimensions. Thus, we kindly hope to keep the current table form.
>
> ---
>
> Thank you once again for your constructive feedback. If you have any questions or concerns, please let us know.

---

> > ### Author Response · Authors · 2025-11-27
> >
> > Dear Reviewer 3MFr,
> >
> > It is our great honor to have you as a reviewer for our paper. We sincerely appreciate the time and effort you have devoted to your review. We have carefully clarified or addressed all the concerns you raised, and sincerely hope that these clarifications and revisions have resolved them.
> >
> > As the discussion phase is nearing its end, we just wanted to gently check whether you have any remaining concerns. We would be very happy to clarify anything further if needed.
> >
> > Thank you again for your thoughtful feedback. Wishing you a lovely day!
> >
> > Warmly,
> >
> > Authors

---

### Official Review · Reviewer_Avsu · 2025-11-04

**Soundness:** 2
**Presentation:** 3
**Contribution:** 2
**Rating:** 4
**Confidence:** 3

**Summary:**

This paper studies the safety alignment problem for LLMs. Specifically, it proposes a method to defend against malicious fine-tuning samples while keeping the model's utility score. What lies in the core the proposed method is to find a small parameter perturbation which discourages moving along harmful gradient direction when optimizing for the utility loss. The method is able to mitigate harmful gradient direction in utility loss gradient while not affecting the utility loss too much. Experimental results demonstrate the effectiveness of the proposed algorithm.

**Strengths:**

1. The paper addresses an important problem in LLM alignment.
2. The paper is well-written and easy to follow.
3. This work offers some valuable insight by showing the correlation between the descend of the utility loss and that of the harmful loss. The proposed method is somewhat intuitive and easy to implement. The reported experimental result is good.

**Weaknesses:**

1. Insights of how and why the proposed method works: Though the construction of $L_{su}$ is somewhat intuitive, it is still unclear how and why the algorithm works well. Specifically, how can the algorithm achieve lower utility loss than the full SFT on utility dataset, while being biased constantly (from the perturbation) in its utility optimization process? If the perturbation is supposed to be very small, how can it lead to substantial decrease in harmfulness?

2. Lacking critical baselines: it might be beneficial for the author to compare with simply adding the safety data (used in the proposed algorithm to compute negative harmful direction) into the utility dataset. The baseline can be interpreted as simply mixing the utility gradient with the negative harmful direction, which is very comparable to the proposed algorithm. This might lead to further insight into the performance of the algorithm.

As a result, I am overall hesitant to give an accept suggestion. However, I am happy to reconsider my recommendation if they are adequately addressed.

**Questions:**

1. In the proposed algorithm, is perturbation applied to the model parameter each iteration? If not, will it be a good/bad idea to apply the perturbation?

2. How will the algorithm perform under no malicious data? I am considering a scenario where this method is just applied to enhance safety in normal utility fine-tuning process.

**Details Of Ethics Concerns:**

no ethics concern.

---

> ### Author Response · Authors · 2025-11-21
> **Response to Reviewer Avsu (Part 1/2)**
>
> Dear Reviewer Avsu,
>
> Thank you for your insightful feedback and for recognizing our work for addressing an important problem in LLM alignment, well-written and easy to follow, and good experimental results. In response to your concerns, we have carefully revised the manuscript and provide detailed responses below.
>
> ---
>
> **Q1**: In the proposed algorithm, is perturbation applied to the model parameter each iteration? If not, will it be a good/bad idea to apply the perturbation?
>
> **A1**: Yes, the perturbation is applied to the model in each iteration. The perturbation process can be referred to Algorithm 1, line 2-4, in page 6, which is applied in each iteration (for $k$ in range($K$), line 1).
>
> ---
>
> **Q2**: How will the algorithm perform under no malicious data? I am considering a scenario where this method is just applied to enhance safety in normal utility fine-tuning process.
>
> **A2**: We apologize for any potential misunderstanding. Actually, our work is exactly designed for fine-tuning normal tasks without malicious data. As outlined in Algorithm 1, SAP is used to fine-tune useful data $D_{useful}$ from normal tasks, while safety data and harmful data are introduced to identify safety-critical directions. In most of our experiments, the fine-tuning task involves a typical utility task such as Alpaca, Samsum, or ChatDoctor. However, SAP can also be used effectively to defend against malicious fine-tuning attacks as a side benefit, as shown in Section 5.3.Nonetheless, the primary goal and intended use of SAP is to improve safety during the normal utility fine-tuning process. We have made this point clearer in page 1:
>
> > Motivated by these observations and analysis, we propose a safety-aware probing (SAP) optimization paradigm that can effectively reduce the safety risks of LLMs after fine-tuning **on normal utility tasks**.
>
> We truly appreciate We truly appreciate your valuable and detailed feedback. If you have any further questions or concerns, please let us know.your valuable and detailed feedback. If you have any further questions or concerns, please let us know.

---

> ### Author Response · Authors · 2025-11-21
> **Response to Reviewer Avsu (Part 2/2)**
>
> **Q3**: Insights of how and why the proposed method works: Though the construction of $L_{su}$ is somewhat intuitive, it is still unclear how and why the algorithm works well. Specifically, how can the algorithm achieve lower utility loss than the full SFT on utility dataset, while being biased constantly (from the perturbation) in its utility optimization process? If the perturbation is supposed to be very small, how can it lead to substantial decrease in harmfulness?
>
> **A3**: Thank you for raising this point. We agree that SAP can achieve even better utility performance (lower utility loss), which is an interesting property. We have intuitively explained this effect in Appendix B.2. Specifically, this phenomenon can be attributed to the benefit of random weight perturbations during optimization. Recall that the original SAM [1] enhances natural performance by perturbing the parameters during optimization. Recently, this intriguing property has been explained by the fact that perturbing the parameter space is equivalent to conducting data augmentation within the parameter space [2,3], and even adding random weight perturbations can enhance the generalization of LLMs during pre-training or fine-tuning [4]. Specifically, ensembling gradients through diverse model parameters in a small region results in more reliable optimization.
>
> The unexpected benefit of SAP in improving utility can also be explained this way. While the perturbation of SAP aims at safety, this perturbation still introduces brittle noise into the parameter space, leading to a similar parameter space augmentation mechanism that slightly improves natural utility compared to SFT.
>
> [1] Sharpness-aware Minimization for Efficiently Improving Generalization, ICLR 2021
>
> [2] On the Duality Between Sharpness-Aware Minimization and Adversarial Training. ICML 2024
>
> [3] A Flat Minima Perspective on Understanding Augmentations and Model Robustness. arXiv:2505.24592
>
> [4] Understanding Pre-training and Fine-tuning from Loss Landscape Perspectives. arXiv:2505.17646
>
> ---
>
> **Q4**: Lacking critical baselines: it might be beneficial for the author to compare with simply adding the safety data (used in the proposed algorithm to compute negative harmful direction) into the utility dataset. The baseline can be interpreted as simply mixing the utility gradient with the negative harmful direction, which is very comparable to the proposed algorithm. This might lead to further insight into the performance of the algorithm.
>
> **A4**: Thank you for your suggestion. We would like to argue that the SafeInstr baseline we compared is exactly the method that simply adds the safety data into the utility set. Please refer to Appendix C.3 for more introduction on all baselines, where the SafeInstr is explained as follows:
>
> > SafeInstr: Following the method described in the paper (Bianchi et al., 2023), we incorporated 3% of safety-related instructions and responses into the fine-tuning data.
>
> According to all results throughout our paper, SAP consistently outperforms SafeInstr in terms of both safety preservation and task-specific utility enhancement.
>
> ---
>
> Thank you once again for your constructive feedback. If you have any questions or concerns, please let us know.

---

> > ### Author Response · Authors · 2025-11-27
> >
> > Dear Reviewer Avsu,
> >
> > It is our great honor to have you as a reviewer for our paper. We sincerely appreciate the time and effort you have devoted to your review. We have carefully clarified or addressed all the concerns you raised, and sincerely hope that these clarifications and revisions have resolved them.
> >
> > As the discussion phase is nearing its end, we just wanted to gently check whether you have any remaining concerns. We would be very happy to clarify anything further if needed.
> >
> > Thank you again for your thoughtful feedback. Wishing you a lovely day!
> >
> > Warmly,
> >
> > Authors

---

### Author Response · Authors · 2025-11-23
**Summary of Revision and Invitation for Further Discussion**

We thank all reviewers again for your constructive comments and valuable suggestions. We have revised our manuscript in accordance with your suggestions. Specifically, we have added/extended extensive supplementary content (e.g., experiments, analyses, visualizations, or comparisons) to further address your concerns and strengthen the validity of our work. Key additions are listed below:

- **Table 2 (extended)**: Address the missing baselines concern by adding Booster and Vaccine comparisons across multiple datasets, demonstrating SAP outperforms them by 4.5–5.57 points in harmful scores while improving utility.
- **Section 5.4 (extended)**: Supplemented “same training time” comparison (Figure 5) across all baselines (not just Lisa), proving SAP outperforms baselines with 5 epochs vs. baselines’ 15 epochs.
- **Figure 1 (revised)**: Improve clarity by supplementing detailed captions that explicitly link the visualization to SAP’s core mechanism (safety-critical direction perturbation and hidden-state probe).
- **Appendix.B.2(new)**: We have intuitively explained why SAP can achieve even better utility performance (lower utility loss) than SFT, which is an interesting property.
- **Appendix B.9 (new)**: Address the scalability concern across fine-tuning paradigms by adding full-parameter fine-tuning experiments on Qwen3-0.6B, showing 14.3-point lower harmful scores than SFT while preserving task performance.
- **Appendix B.10 (new)**: Address the scalability concern on larger models by adding LoRA fine-tuning experiments on Llama2-13B, achieving 12.3-point lower harmful scores than SFT with comparable BRT and CL metrics.
- **Appendix B.11 (new)**: Respond to computational cost concerns by supplementing time/memory overhead analysis on Llama2-13B and Qwen-0.6B, showing SAP’s overhead remains ~2.5x that of SFT (no extra GPU memory) and better efficiency under equal training time.
- **Figures 2 & 3 (revised captions)**: Address the lack of dataset/notation clarity by updating captions: 1) Figure 2 caption explicitly states “we apply CircuitBreaker (harmful) (Zou et al., 2024) and Alpaca (useful) (Taori et al., 2023) datasets”; 2) Figure 3 caption clarifies useful-critical direction refers to Definition 3.2, Equation 5, and the harmful-critical direction with datasets consistent with Figure 2.

These revisions and supplements adequately address your concerns and are supported by concrete experiments or explanations. We welcome further questions or insights to refine our work and look forward to your continued guidance. Thank you again for your expertise.

---

### Author Response · Authors · 2025-11-29
**Final Remarks for Area Chair**

Dear Area Chair,

We sincerely appreciate you stepping in to oversee the review process for our submission under these unusual circumstances. We understand the significant additional workload this reassignment entails and are grateful for your time and effort in evaluating our work, the original reviews, and our rebuttal updates.

---

**Summary of Strengths**

The reviewers have highlighted the following key strengths of our work:

- **Effective & Principled Method:** The proposed Safety-Aware Probing (SAP) is simple, principled, and effectively steers updates away from harmful directions while preserving (or improving) task utility (Reviewers Avsu, 3MFr, PBzG).
- **Strong Empirical Results:** The method demonstrates effectiveness across diverse settings, including benign fine-tuning, poisoning, and adversarial attacks (Reviewers 3MFr, PBzG).
- **Comprehensive Evaluation:** Experiments are extensive, covering three models, instruction-following/reasoning benchmarks, and various attack scenarios (Reviewer kkBd).
- **Novel Insight:** The paper offers valuable insight into the correlation between utility loss descent and harmful loss descent (Reviewer Avsu).

---

**Response to Key Concerns**

**1. Scalability and Generalization**

- **Concern:** The claim of scalability was not initially supported by experiments on larger models or full-parameter fine-tuning (Reviewer PBzG).
- **Resolution:** We addressed this by conducting **new experiments** during the rebuttal phase.
    - **Full-Parameter Fine-Tuning:** We applied SAP to Qwen3-0.6B, achieving a 14.3-point reduction in harmful scores compared to SFT while maintaining utility.
    - **Larger Models:** We applied SAP to Llama2-13B (LoRA), achieving a 12.3-point reduction in harmful scores with comparable utility.
- **Location:** (**New Revision)** Appendix B.9 and Appendix B.10.

**2. Missing Baselines**

- **Concern:** Reviewers requested comparisons to `Booster` and `Vaccine` beyond Table 1 (Reviewer 3MFr) and a baseline that simply mixes safety data into the utility dataset (Reviewer Avsu).
- **Resolution:**
    - **For `Booster`/`Vaccine`:** We first **clarified** why we only compared Booster and Vaccine in Table 1: they are designed for general alignment enhancement, not for preserving safety during task-specific fine-tuning like our SAP. To make this point clearer, we have explained this difference in our experiment setup, and **added additionally comparisons** to the manuscript in Table 2, where SAP outperforms them by 4.5–5.5 points in harmful scores.
    - **For Mixing Safety Data:** We **clarified** that the existing baseline `SafeInstr` represents exactly this approach (incorporating safety data into the utility dataset). SAP consistently outperforms `SafeInstr` (HS 23.1% vs 26.2%).
- **Location:** **(New Revision)** Table 2 (Extended Results); **Original Content**: Appendix C.3 (Baseline Details).

**3. Computational Cost and Efficiency**

- **Concern:** SAP introduces a 2x-3x time overhead per batch; detailed comparison under equal time budgets were needed (Reviewers PBzG, kkBd).
- **Resolution:** We added a "Same Training Time" analysis for SAP and all baselines, where we demonstrated that SAP trained for just 5 epochs significantly outperforms all baselines trained for 15 epochs, indicating that SAP achieves better performance with even less training time.
- **Location:** **(New Revision)** Section 5.4 (Figure 5) and Appendix B.11 (Table 16).

**4. Mechanism Clarification (Utility Improvement)**

- **Concern:** It was unclear why SAP could achieve lower utility loss than SFT despite introducing perturbations (Reviewer Avsu).
- **Resolution:** We added a discussion explaining this via the lens of parameter space augmentation (similar to Sharpness-Aware Minimization). The perturbation acts as a regularizer that smooths the landscape, aiding generalization.
- **Location:** **(New Revision)** Appendix B.2.

**5. Hyperparameter Complexity**

- **Concern:** The method seemed to introduce too many hyperparameters ($\alpha, \beta, \epsilon$) (Reviewer 3MFr).
- **Resolution:**
    - **Clarification:** We clarified that $\alpha$ is the standard learning rate and $\epsilon$ is fixed.
    - **New Variant:** We proposed an **Adaptive SAP** variant that automatically calibrates the probe update step $\beta$. Results show Adaptive SAP matches the performance of the fixed version, reducing tuning complexity.
- **Location:** **(Original Content)** Appendix B.5.

**6. Adaptive Attacks**

- **Concern:** The evaluation needed to consider adaptive attackers of SAP (Reviewer kkBd).
- **Resolution:** We **clarified** that these experiments were already present in the manuscript. We tested Pre- and Post-Fine-Tuning adaptive attacks where the adversary targets the safety probe. SAP maintains robust defense (Harmful Score <28) compared to SFT (Harmful Score 60) in these scenarios.
- **Location:** **(Original Content)** Appendix B.4.

---

### Meta-Review · Area_Chair_5Jcp · 2025-12-03

**Summary:**

The paper received mixed reviews, with three reviewers rating it 4 (marginally below acceptance) and one rating it 6 (marginally above acceptance). Reviewers consistently praised the Safety-Aware Probing (SAP) method for being simple, principled, and effective at mitigating safety risks while maintaining task utility. However, significant concerns were raised regarding the method's computational cost (introducing a 2x-3x time overhead), the initial lack of evidence for scalability to larger models or full-parameter fine-tuning, and missing baseline comparisons. Critics also requested deeper theoretical insights into why the method improves utility despite perturbations and noted the complexity introduced by its multiple hyperparameters.

I thank the authors for all the effort they have put into this work. After careful consideration, I decided against accepting the paper. While I share some of the sentiments raised by the reviewers, after thoroughly reading the paper I remain unconvinced that the proposed approach is well justified. In particular, it is not clear why the seemingly sophisticated method works, and there are many other approaches that would be more natural to try first, gradually increasing in complexity before arriving at the proposed formulation.

My first instinct is to question the necessity of introducing perturbations in the feature space, referred to as probing in the paper. A more intuitive starting point would be to consider the simpler objective $\max_W \mathcal{L}_{\text{su}}$ which naturally seeks parameters that incur high usefulness-related loss around parameters that correspond to harmful data, while simultaneously minimizing the loss on usefulness data. This seems to be the most direct and interpretable approach. Only if such a baseline fails—and we understand clearly why it fails—would it be justified to introduce more complex mechanisms such as probing. At present, the paper does not make sufficiently clear why probing is needed or what precise benefit it brings.

After all, perturbing the activations in the proposed manner (i.e., the lower-level problem $\max_V \mathcal{L}_{\text{su}} (W,V)$) merely ensures that the parameters W obtained by Algorithm 1 perform well in minimizing the usefulness loss even under the “best-case” probes, i.e., activations most favorable to usefulness and distinct from harmful directions. It remains unclear why this provides a meaningful advantage over simpler adversarial or robustness-inspired formulations. Potentially even under probes that are adversarial in natura, i.e., working towards improving the usefulness in the harmful direction.

Overall, the methodology requires stronger motivation and more extensive empirical evidence. The claim that the optimal solution for a usefulness objective shares its solution space with one that is harmful is interesting (Figures 2/3), but it does not convincingly motivate the need for probing or the complexity of the proposed objective. This seems to have come out of the blue and it is utterly unclear to me how is probing helpful if at all or whether it is someother part of the proposed bigger loss function that is contributing to the problem. A more principled development, starting from simpler adversarial losses and progressively increasing complexity until the proposed appraoch, would help clarify why the proposed approach is necessary and preferable.

**Reviewer Concerns:**

see above

**Reviewer Scores:**

see above

---

### Decision · Program_Chairs · 2026-01-26

Reject